# The North Italian Longitudinal Study Assessing the Mental Health Effects of SARS-CoV-2 Pandemic Health Care Workers—Part II: Structural Validity of Scales Assessing Mental Health

**DOI:** 10.3390/ijerph19159541

**Published:** 2022-08-03

**Authors:** Emanuele Maria Giusti, Giovanni Veronesi, Camilla Callegari, Gianluca Castelnuovo, Licia Iacoviello, Marco Mario Ferrario

**Affiliations:** 1Psychology Research Laboratory, Istituto Auxologico Italiano IRCCS, 20149 Milan, Italy; e.giusti@auxologico.it; 2Department of Psychology, Catholic University of the Sacred Heart, 20123 Milan, Italy; gianluca.castelnuovo@unicatt.it; 3EPIMED Research Center, Department of Medicine and Surgery, University of Insubria, 21100 Varese, Italy; giovanni.veronesi@uninsubria.it (G.V.); licia.iacoviello@uninsubria.it (L.I.); 4Division of Psychiatry, Department of Medicine and Surgery, University of Insubria, 21100 Varese, Italy; camilla.callegari@uninsubria.it; 5Psychology Research Laboratory, Istituto Auxologico Italiano IRCCS, 28824 Verbania, Italy; 6Department of Epidemiology and Prevention, IRCCS Neuromed, 86077 Pozzilli, Italy

**Keywords:** factor structure, health care workers, Maslach Burnout Inventory, General Health Questionnaire, PTSD Checklist for DSM-5, Connor-Davidson Resilience Scale, Post-Traumatic Growth Inventory, mental health, COVID-19, longitudinal study

## Abstract

It is unclear if the factor structure of the questionnaires that were employed by studies addressing the impact of COVID-19 on the mental health of Healthcare Workers (HCW) did not change due to the pandemic. The aim of this study is to assess the factor structure and longitudinal measurement invariance of the Maslach Burnout Inventory (MBI) and the factor structure of the General Health Questionnare-12 (GHQ-12), PTSD Checklist for DSM-5-Short Form (PCL-5-SF), Connor-Davidson Resilience Scale-10 (CD-RISC-10) and Post-Traumatic Growth Inventory-Short Form (PTGI-SF). Out of n = 805 HCWs from a University hospital who responded to a pre-COVID-19 survey, n = 431 were re-assessed after the COVID-19 outbreak. A Confirmatory Factor Analysis (CFA) on the MBI showed adequate fit and good internal consistency only after removal of items 2, 6, 12 and 16. The assumptions of configural and metric longitudinal invariance were met, whereas scalar longitudinal invariance did not hold. CFAs and exploratory bifactor analyses performed using data from the second wave confirmed that the GHQ-12, the PCL-5-SF, the PTGI-SF and the CD-RISC-10 were unidimensional. In conclusion, we found support for a refined version of the MBI. The comparison of mean MBI values in HCWs before and after the pandemic should be interpreted with caution.

## 1. Introduction

The mental health of Healthcare Workers (HCW) is of paramount importance since its reduction is strictly linked to high direct and indirect individual and social costs [1]. Declining mental health in the workplace in general is known to be associated with loss of productivity, higher spending on social security programs and to cause direct expenditure on healthcare [2]. HCWs are at particular risk since they are exposed to a multitude of stressors potentially affecting their physical and mental health [1]. As a consequence, they experience high levels of anxiety, depression, burnout and, due to repeated exposure to patients who have experienced traumas, forms of vicarious traumatization that might give rise to symptoms of Post-Traumatic Stress Disorder (PTSD) [3,4]. The impact of these conditions is particularly of concern, since it might result in shortage of crucial professionals, reduced quality of patient care and medical errors [5].

Before the pandemic, it was estimated that about 11% of nurses and from 20% to 40% of physicians suffered from burnout [6,7]. The COVID-19 pandemic seems to have exacerbated this issue due to repeated exposure to life-threatening situations, fear of contagion, shift overload and changes in work organization [8,9]. The results of longitudinal studies and the comparison between the estimates of the studies performed before and during the pandemic show a high prevalence and a marked increase of the levels of anxiety, with estimates from studies performed during the pandemic ranging from 22% to 31%, depression (from 17% to 36%), PTSD symptoms (from 13% to 37%) and burnout (from 36% to 52%) [10,11,12,13,14,15,16,17]. Among the protective factors, studies have shown that resilience and post-traumatic growth helped to buffer the impact of the pandemic, suggesting that these factors should be taken into account in order to describe the mental health of HCWs [18,19].

However, a caveat applies to this evidence. Both the direct and indirect comparisons across time of the mental health levels are valid if the assumption of the longitudinal measurement invariance of the questionnaires that are used to assess them holds, that is to say that the questionnaires have the same meaning for people over time. If this assumption does not hold, differences in scores across time do not reflect differences in the latent construct of interest but to changes in how people respond to the questionnaire [20]. As a consequence, conclusions about changes in the construct of interest are not valid [21]. Given the enormous psychological, social and cultural impact of the pandemic on the general population, and the specific challenges that HCWs had to face during the pandemic [22], it is uncertain whether the assumption of the measurement invariance and the validity of factor structures analyzed before the pandemic holds. This has important consequences for both the assessment of changes in constructs in longitudinal studies, and for the interpretation of results in cross-sectional studies performed after the pandemic. Therefore, studies on the structural validity of questionnaires employed to assess the mental health of HCWs are needed.

This study is part of a broader research project which aimed to assess the mental health status of HCWs during the pandemic, the changes in burnout from the pre-pandemic to the pandemic period, and the impact of pre-pandemic work-related stressors on mental health during the pandemic. Based on previous literature, the mental health variables that were studied were general mental health, burnout, PTSD symptoms, post traumatic growth and resilience. In a parallel paper, the issues of the representativeness of the sample and of the structural validity of the questionnaires assessing pre-pandemic work-related stressors and work satisfaction are analyzed. In this paper, we aimed to investigate the factor structure of widely used questionnaires assessing the mental health constructs named above. In particular, we assessed the factor structure and the longitudinal measurement invariance of the Maslach Burnout Inventory (MBI) using both pre-pandemic data and data collected during the pandemic, as well as the factor structure of the validated Italian versions of the General Health Questionnaire-12 (GHQ-12), the PTSD Checklist for DSM-5-Short Form (PCL-5-SF), the Connor-Davidson Resilience Scale-10 (CD-RISC-10) and the Post-Traumatic Growth Inventory-Short Form (PTGI-SF), using data collected during the pandemic.

## 2. Materials and Methods

From August to September 2019 all the doctors, nurses, assistant nurses and office clerks working in 69 selected wards and offices of a large University hospital in the city of Varese participated in a first online survey investigating work-related stress and mental health variables (from now on: pre-COVID-19 wave). The wards and offices were selected to include those at expected high risk of work stress according to the representatives of the unions and of the hospital administration, and a random sample of the remaining wards. Then, from December 2020 to January 2021, the respondents to the first survey were asked to participate in a second online survey investigating the impact of the COVID-19 pandemic on their mental health (from now on: COVID-19 wave). Additional details regarding the study population, design and procedures are provided in the companion paper [23]. The study received approval from the Institutional Ethics Committee (approval ID 69/2020).

### 2.1. Measurement Instruments

The following measurement instrument was administered both during the first and second survey:Maslach Burnout Inventory (MBI) [24]. The MBI is a self-report questionnaire which assesses the three theoretical components of burnout, i.e., emotional exhaustion, depersonalization and personal accomplishment. It is composed of 22 Likert-type items assessing the frequency of feelings or attitudes reflecting the three components of burnout on a scale ranging from 0 (“never”) to 6 (“every day”). Higher scores in the emotional exhaustion and depersonalization scales reflect higher levels of burnout, whereas higher scores in the personal accomplishment subscale reflect lower levels of burnout. Reviews on the psychometric properties of the MBI showed that this measurement instruments has adequate construct validity and internal consistency [25]. The Italian translation of the MBI was found to reflect the same three-factor structure as the original version and showed adequate internal consistency [26].

The following measurement instruments were administered only in the second survey:General Health Questionnaire-12 (GHQ-12) [27]. The GHQ-12 is a self-report questionnaire originally developed for use in consulting settings to detect individuals with psychiatric disorders, whose 12-items version is an extensively used general measure of mental well-being. The GHQ-12 items assess the severity of a mental problem over the past two weeks using a 4-point Likert scale ranging from 0 (“less than usual”) to 3 (“much more than usual”). Higher scores indicate worse mental well-being. The factor structure of the GHQ-12 is debated, since unidimensional and multidimensional structures have been proposed [28,29,30]. However, it has been recognized that the retrieval of multiple dimensions is related to a wording effect rather than the presence of true distinct latent factors, since half of the GHQ-12 items have a positive wording and half a negative wording [31]. The Italian version of the GHQ-12 showed acceptable test–retest reliability [32].The PTSD Checklist for DSM-5-Short Form (PCL-5-SF) [33]. The PCL-5-SF is a 5-item version of the original 20-item PCL-5 questionnaire, which is a self-report measure assessing the frequency of the DSM-5 symptoms of PTSD in the past month using a Likert scale ranging from 1 (“Not at all”) to 5 (“extremely”). Higher scores indicate higher frequency of PTSD symptoms. The Italian version of this questionnaire has shown adequate construct validity, criterion validity and internal consistency [34].The Connor-Davidson Resilience Scale-10 item version (CD-RISC-10) [35]. The CD-RISC-10 is a self-report questionnaire assessing resilience based on the Connor and Davidson definition, i.e., the ability to thrive in the face of adversity [35]. In this study, we employed the 10-item version. The items are scored using Likert scale ranging from 0 (“not true at all”) to 4 (“true nearly all the time”). Higher scores indicate higher levels of resilience. The Italian version of the CD-RISC-10 has shown adequate internal consistency and concurrent validity [36].The Post-Traumatic Growth Inventory-Short Form (PTGI-SF) [37]. The PTGI-SF is a self-rated questionnaire assessing positive outcomes reported by people who have experienced traumatic events. In this study we employed the 10-item version of this scale. The items are scored on a 6-point Likert scale ranging from 0 (“I did not experience this change as a result of my crisis”) to 5 (“I experienced this change to a very great degree as a result of my crisis”). Higher scores indicate higher levels of post-traumatic growth. The Italian version of the PTGI-SF has shown adequate internal consistency [38].

### 2.2. Statistical Analyses

#### 2.2.1. Preliminary Analyses

Demographic and work-related variables were described using frequencies and percentages if categorical, and using means and standard deviations if continuous. Then, a missing data analysis was performed. Firstly, the absolute and relative amount of missing data for each measurement instrument was inspected. Then, patterns of missing data were inspected by plotting the frequency of each missing data pattern. The Little’s test was finally performed to assess if the missing data mechanism was Missing Completely At Random (MCAR) [39]. This test was run twice, once using the variables collected in the first survey, namely the sociodemographic variables and the MBI data, and once using the sociodemographic variables and the variables collected in the second survey. Then, the normality of each questionnaire was checked, performing both a Henze-Zirkler test for multivariate normality and, for each item, inspecting normal Q-Q plots and the results of Shapiro-Wilk tests.

#### 2.2.2. Confirmatory Factor Analyses

Measurement instruments whose items are scored using Likert scales with less than 5 points were analyzed using a Diagonally Weighted Least Squares (DWLS) estimator, and those with more than 5 points using a Maximum Likelihood estimator with robust standard error and a Satorra-Bentler scaled test statistic (MLR). The use of the robust variant of the maximum likelihood estimator was chosen because the normality assumption of all the questionnaires was violated.

All the CFAs were performed by assessing the fit of the factor structures already analyzed by previous studies. In case of poor fit of all these factor structures, items were deleted one by one by combining theoretical considerations (e.g., presence of items that were flagged as problematic in other studies either for content issues or for displaying bad fit with the proposed structures) and the assessment of misfit through modification indices, which indicate presence of cross-loadings or of correlated errors between items. This procedure was performed iteratively until a structure with good fit was reached. Cut-offs for good fit were CFI ≥ 0.95 and RMSEA ≤ 0.06 for CFAs that were estimated using the MLR estimator and AGFI ≥ 0.90 and SRMR ≤ 0.08 for CFAs that were estimated using the DWLS estimator [40]. Structures were retained also when the fit indices were CFI ≥ 0.90 and RMSEA ≤ 0.08 [41] and further deletion of items had no theoretical significance.

#### 2.2.3. Confirmatory Factor Analyses and Measurement Invariance of the Maslach Burnout Inventory

Firstly, the factor structure of the MBI was analyzed using the data from the whole sample during the pre-COVID-19 wave. The first structure that was evaluated was the three correlated factors structure employed by Sirigatti & Stefanile [26] for the Italian translation of the MBI, which is identical with the structure proposed by Maslach et al. [24]. Then, items were deleted following the procedure described above.

Then, the longitudinal invariance of the MBI across the waves was assessed using the data of HCWs who responded to both the pre-COVID-19 and the COVID-19 surveys. This was done by comparing the fit of models, progressively posing stricter equality constraints between the responses of each HCW at the two waves. Firstly, configural invariance was tested by constraining the items to load on the same latent factors in both waves. Secondly, metric invariance was tested by constraining item loadings to be equal across the waves. Finally, scalar invariance was tested by constraining item intercepts to be equal. The presence of each of these components of measurement invariance was established if the difference in CFI (ΔCFI) and RMSEA (ΔRMSEA) between the more constrained structure and the preceding structure was lower than 0.01 and 0.015, respectively [42]. The errors of each item across time were allowed to correlate. 

#### 2.2.4. Factor Analyses of the GHQ-12, of the PCL-5-SF, of the PTGI-SF and of the CD-RISC-10 Scale

The factor structure of the GHQ-12, of the PCL-5-SF, of the PTGI-SF and of the CD-RISC-10 scale was assessed using the data from the second wave. A unidimensional structure was firstly assessed using a confirmatory approach. If available, other factor structures based on previous studies were assessed. If no adequate factor structures were found, we performed a bifactor analysis to assess if the measurement instrument under consideration was “unidimensional enough” to allow for the calculation of a total score [43]. Briefly, in a bifactor structure, items are assumed to reflect both a general factor that accounts for covariation among items, and a specific factor that accounts for covariation not measured by general factor. Based on the resulting bifactor structure, it is possible to calculate the Explained Common Variance (ECV), which is an estimate of the amount of variance that is shared among a set of items, and the coefficient omega-hierarchical (ωH), which is an estimate of the general factor saturation of a test. An ECV > 0.70 and a ωH > 0.80 support the presence of a strong general factor [44]. The bifactor structures were assessed using a confirmatory approach. In case of non-convergence of the confirmatory bifactor structure, an exploratory bifactor analysis was performed using the Schmid-Leiman procedure [45].

Finally, the Cronbach’s alpha of the final versions of the questionnaires was computed. All the analyses were performed using the significance value of 0.05. The CFAs were performed using SAS OnDemand for Academics software (release 9.04) (SAS Institute Inc., Cary, NC, USA). The exploratory bifactor analysis was performed using the R (version 4.0.1) package psych [46].

## 3. Results

Of the 1286 HCWs who were invited, 805 participated in the pre-COVID-19 wave. The socio-demographic and work-related characteristics of the sample are reported in Table 1. Four hundred and thirty-one of them also responded to the COVID-19 wave. The flowchart of the participants is reported in Appendix A (Figure A1). 

### 3.1. Missing Data Analysis

The percentage of cases with at least one missing value in the MBI items was 8.07%. The inspection of missing data patterns did not reveal any potential recurring pattern. The Little’s MCAR test performed using the data from the first survey was not significant (χ^2^(916) = 947.83, *p* = 0.22). Since missing data were less than 10% and the hypothesis of presence of MAR mechanisms was not corroborated, observations with missing data were excluded from the analysis of the factor structure of the MBI. Missing data of the questionnaires collected during the second survey were less than 1% for the GHQ-12, 1.6% for the PCL-5-SF, 2.7% for the PTGI-SF and 2.6% for the CD-RISC-10. The Little’s MCAR test performed using the data from the second survey was significant (χ^2^(1075) = 1186.32, *p* = 0.01). Nonetheless, the graphical analysis of the missing data revealed no pattern of missing data. Based on this finding and on the low amount of missing data, the CFAs of each of the questionnaires administered during the second wave were performed after excluding cases with missing observations.

### 3.2. Confirmatory Factor Analysis and Internal Consistency of the Maslach Burnout Inventory

The results of the CFA performed to assess the factor structure of the MBI are summarized in Table 2. The fit of the original three-correlated factors structure was not satisfactory. Based on presence of cross-loadings and the fact that these items were flagged as misfitting by previous studies, items 12 and 16 were excluded from the structure and the CFA was re-run. The fit of this solution was not satisfactory. Based on previous literature and presence of correlated errors between items 2 and 6, items 2 and 1 and items 2 and 3, we decided to remove item 2. The fit of the resulting structure did not reach the criteria. Therefore, we decided to remove item 6 due to presence of cross-loadings, correlated errors with item 5 and presence of misfit in previous studies. The resulting refined version of the MBI reached the cut-off criteria. This version was therefore employed in the analyses of the measurement invariance. The loadings of this solution are represented in the Appendix A (Figure A2). The Cronbach’s alphas of the subscales of the refined version of the MBI were above the cutoff (Emotional Exhaustion: alpha = 0.89, Depersonalization: alpha = 0.72, Personal Accomplishment: alpha = 0.74).

### 3.3. Longitudinal Measurement Invariance of the Refined Version of the Maslach Burnout Inventory

The results of the longitudinal measurement invariance are reported in Table 3. The configural and metric invariance of the MBI over time was corroborated. However, the scalar invariance assumption did not hold. After relaxing the equality constraints of the intercepts of five items based on their modification indices, the fit indices did not reach the pre-defined criteria. Therefore, the scalar invariance was not corroborated. The analysis of the sign of the mean differences in MBI scores, given the same amount of the latent trait, suggests that post-pandemic scores are higher (data not presented). 

### 3.4. Factor Analysis and Internal Consistency of the General Health Questionnaire-12, of the PCL-5-SF, of the Post-Traumatic Growth Inventory and of the Connor-Davidson Resilience Scale 

Regarding the GHQ-12, the fit of the unidimensional structure was satisfactory (AGFI = 0.97, SRMR = 0.08). We also assessed the fit of a bidimensional structure based on the wording of the items, in which items with positive wording loaded on a positive factor and items with negative wording loaded on a negative factor. This factor structure showed slightly better indices (AGFI = 0.98, SRMR = 0.07) with respect to the unidimensional structure. Since the fit of the unidimensional structure was satisfactory and theoretically sounder than the bidimensional structure, we decided to retain the unidimensional structure. The loadings of this structure are reported in the Appendix A (Figure A3). The Cronbach’s alpha of the scale was 0.88.

The fit of the unidimensional structure applied to the PCL-5-SF did not meet the pre-specified cutoffs (CFI = 0.98, RMSEA = 0.12). The Schmid-Leiman procedure did not converge. Since no other plausible factor structures were found in the literature and there was no theoretical background for the deletion of items, we did not attempt to perform additional analyses. The Cronbach’s alpha of the scale was 0.84.

The unidimensional CFA performed on the PTGI-SF did not reach the cutoff criteria (CFI = 0.92, RMSEA = 0.10). We tested a 5-factor structure based on the solution of the original version of the scale [49], but the covariance matrix based on this structure was not positively definite. We found no theoretical background for removal of individual items. Therefore, we performed an exploratory bifactor analysis using the Schmid Leiman procedure. The values of ωH and ECV calculated using this procedure were above the cutoffs (ωH = 0.80, ECV = 0.74), providing support for the calculation of a total score. The loadings of the exploratory bifactor analyses are reported in the Appendix A (Figure A4). The Cronbach’s alpha of the scale was 0.92.

Finally, the CFA performed on the CD-RISC-10 showed that the unidimensional structure did not reach the cutoff criteria (CFI = 0.94, RMSEA = 0.07). We found no alternative factor structures in the literature, nor any theoretical background for the removal of any item. Therefore, we performed an exploratory bifactor analysis of the scale. The bifactor indices were both above the cutoffs (ωH = 0.80, ECV = 0.74). The loadings of this solution are reported in Appendix A (Figure A5). The Cronbach’s alpha of the scale was 0.89.

## 4. Discussion

The aim of this study was to assess the factor structure and measurement invariance of measurement instruments of interest for the assessment of mental health in HCWs. The results show that a refined version of the MBI has adequate structural validity and longitudinal measurement invariance over the pandemic waves, that the GHQ-12 is a unidimensional scale, and that factor structure of the PCL-5-SF, the PTGI-SF and the CD-RISC-10 is uncertain but that these questionnaires can be scored as unidimensional instruments. 

The factor structure of the MBI has been widely studied in different countries and work contexts [50,51,52]. The results of these studies are often conflicting, and this has led to a debate about the internal validity of the scale, with various authors noting that the MBI might have different characteristics in different populations, and various studies have suggested some modifications to the scale [51,53,54]. In most of these studies, the presence of three correlated factors reflecting the components of burnout is corroborated, but there are differences in the retained items. Loera et al. [50] performed multiple CFAs using data from Italian nurses evaluating the fit of ten factor structures analyzed by previous studies, finding support for the deletion of items 2, 12 and 16. The results of this study provide additional support for the removal of these items, with the addition of the removal of item 6. In our sample, removing these items helped to find a factor structure with good fit indices, and the resulting questionnaire showed adequate measurement invariance over time. 

The misfit of items 12 (“I feel very energetic”) and item 16 (“Working with people directly puts too much stress on me”) has been previously recognized by Maslach herself [55] and is probably due to their content. The content of item 12 could refer both to absence of emotional exhaustion and presence of personal accomplishment, and the content of item 16 might be both a symptom of emotional exhaustion and, since it refers to difficulties in being in contact with people, might overlap with depersonalization. Item 2 has been recognized as misfitting by a previous study [47] and its deletion improved the fit on the study by Loera et al. [50]. In our study, we found that this item has cross-loadings with the components of emotional exhaustion and depersonalization and that its errors are correlated with those of items 1 (“I feel emotionally drained from my work”) and 3 (“I feel fatigued when I get up in the morning and have to face another day on the job”). The presence of cross-loadings could not be explained by content issues, since it refers to a symptom of emotional exhaustion (“I feel used up at the end of the workday”) that should not be related to depersonalization. However, the fact that its errors are correlated with those of three items of the same scale referring to fatigue seems to suggest that this item causes redundancy. Finally, item 6 (“Working with people all day is really a strain for me”) seems to suffer from the same issue as item 16, since it refers both to emotional exhaustion and difficulties in relationships with people, an aspect that might reflect depersonalization. The deletion of this item, which similarly to items 2 and 16 should reflect emotional exhaustion, improved the fit while still leaving a sufficient number of items in that factor, which preserves its content validity. The 18-items refined version of the MBI showed adequate structural validity and adequate internal consistency. 

The refined version of the MBI meets the requirement for configural and metric longitudinal invariance, but not for scalar invariance. Presence of configural invariance suggests that item responses reflect the same latent factors across waves, meaning that responders at different waves have equal interpretations of the latent factors. Presence of metric invariance suggests that the relationships between the responses to the measurement instrument and the latent variable are similar over time. Violation of scalar invariance suggests that, given the same level of the latent trait, scores obtained during the pandemic are higher than scores obtained before the pandemic. Overall, these results provide support for the use of this refined version of the MBI for the assessment of burnout both in the pre-pandemic and in the post-pandemic period. However, the assessment of changes in burnout levels between the pre-pandemic and the post-pandemic period should be made with caution since differences can be attributable to different ways of scoring the questionnaire. 

The absence of scalar invariance of the refined version MBI over the pandemic waves has some consequences. Cross-sectional studies have shown that burnout after the pandemic has a very high prevalence, and longitudinal studies that burnout levels have increased as a result of the pandemic [8,10,11,12,13,14,15,16,17]. If the lack of longitudinal scalar invariance that was found in this study is generalized to other countries and populations, it might be hard to disentangle the presence of true differences in the levels of burnout from differences in how HCWs respond to the items of the MBI from the pandemic on. This calls for further research to confirm the longitudinal measurement invariance of the MBI, possibly using advanced psychometric methods that might also help to provide information about the magnitude of its impact [56]. In addition, the stability of the cut-offs employed to detect presence of clinically significant levels of burnout should be specifically assessed. Given the absence of studies supporting the longitudinal invariance of the MBI, we suggest explicitly taking this issue into account. Methods for taking the absence of invariance into account have been proposed, and vary according to the type of study [57,58,59,60]. In any case, the comparison of burnout over time and of change scores should be interpreted with caution.

The analysis of the GHQ-12 showed that a unidimensional structure had a good fit to the data. The factor structure of the GHQ-12 has been debated but it has been argued that the retrieval of distinct latent factors is an artifact due to the positive and negative wording of the GHQ-12 items [31]. In this study, we found support for a slightly better fit of the bidimensional structure accounting for the wording effect. Nonetheless, since the unidimensional structure is in line with the theoretical structure underpinning the measurement instrument [30], we suggest consideration of the GHQ-12 as a unidimensional scale.

When we analyzed the factor structure of the PCL-5-SF, we found no support for the structural validity of this measurement instrument. Some caveats must be made. Firstly, from a statistical point of view, analyzing a scale composed of four items has the consequence that the fit indices, in particular the RMSEA, tend to detect misfit even in its absence [61]. Secondly, the PCL-5-SF was developed from the original measurement instrument using the maintenance of diagnostic ability as a criterion rather than focusing on its psychometric properties, and in such a way as to be consistent with DSM-V criteria [62]. Therefore, modifications of the scale are not warranted, and the absence of a good fit of a one-dimensional structure does not preclude its use.

Finally, both the CD-RISC-10 and, in particular, the PTGI-SF showed some departure from unidimensional factor structures. The PTGI-SF has been studied either as a unidimensional instrument or as a 5-factor instrument [49,63,64]. In this study, we found that both solutions did not reach a good fit to the data. However, the exploratory bifactor analysis provide some ground for the calculation of a total score. Similarly, the CD-RISC-10 has been described as a unidimensional scale [65,66], but the CFA showed that the fit of the unidimensional structure was not satisfactory. In this case also, the exploratory bifactor analysis showed the presence of a strong general factor, supporting the calculation of a total score. It is important to note that the results of the exploratory bifactor analysis should be analyzed with caution and should not be employed as a strong corroboration for the unidimensionality of these scales [43]. Analyses using advanced psychometric methods are warranted to gather insights about the factor structure of these scales [44]. 

There are various aspects that might impact the generalization of the results of this study. Firstly, the second survey was performed at a specific time within a period characterized by frequent and sudden changes in the characteristics of the pandemic and the burden on the healthcare system. In particular, the survey was performed in a period when the pandemic was at the beginning of its second wave and vaccines were not already available. It is therefore possible that the results of this study, in particular the results of the longitudinal measurement invariance, could have been different if the survey had been carried out at a different time. In addition, given the territorial variability in the effect if the pandemic burden on hospitals, the results of our study are not generalizable to contexts in which the impact of the pandemic was less severe. 

Our study has some limitations. Firstly, we did not perform a formal analysis to determine the initial sample size. Therefore, we may lack sufficient precision for subgroup analysis by job title, especially in the COVID-19 wave. In this paper, for instance, we were not able to assess the measurement invariance of the questionnaires across HCW groups. Secondly, except for the MBI, the questionnaires analyzed in this study were administered only in the COVID-19 wave and we could not perform the assessment of longitudinal invariance of these instruments. Finally, the presence of selection bias due to minor rates of responses of workers with high levels of burnout in both waves cannot be excluded. Nonetheless, at least for work-related characteristics, the representativeness of the sample has been confirmed [23] and it is unlikely that this issue might have impacted our structural validity assessments.

## 5. Conclusions

In conclusion, we found support for the use of an 18-item refined version of the MBI for the assessment of burnout in Italian HCWs and we suggest that the GHQ-12, the PCL-5-SF, the PTGI-SF and the CD-RISC-10 should be scored as unidimensional scales. The comparison between pre-pandemic and post-pandemic levels of burnout should be made and interpreted with caution.

## Figures and Tables

**Table 1 ijerph-19-09541-t001:** Description of the participants in the pre-COVID-19 wave (n = 805).

Variable	Level	n (%)	Missing Data n (%)
Sex	Female	622 (77.6)	3 (0.3)
	Male	180 (22.4)	
Age	25–34 years	126 (15.7)	1 (0.1)
	35–44 years	178 (22.1)	
	45–54 years	306 (38.1)	
	55 years or more	194 (24.1)	
Job profile	Clerk	102 (12.7)	
	Doctor	104 (12.9)	
	Nurse	429 (53.3)	
	Nurse assistant	170 (21.1)	
Education attainment	Primary school	115 (14.3)	1 (0.1)
	Secondary school	295 (36.7)	
	Attended university	20 (2.5)	
	Bachelor’s degree	192 (23.9)	
	Master’s degree	61 (7.6)	
	Postgraduate programs or specialization	121 (15.0)	
Job seniority	Less than 1 year	57 (7.1)	3 (0.3)
	From 1 to 5 years	167 (20.8)	
	From 6 to 15 years	190 (23.7)	
	From 16 to 30 years	255 (31.8)	
	More than 30 years	133 (16.6)	
Work seniority in current role	Less than 1 year	62 (7.7)	2 (0.2)
	From 1 to 5 years	232 (28.9)	
	From 6 to 15 years	223 (27.8)	
	From 16 to 30 years	230 (28.6)	
	More than 30 years	56 (7.0)	
Type of contract	Fixed-term	14 (1.7)	2 (0.2)
	Permanent	789 (98.3)	
Type of employment	Full-time	693 (86.3)	2 (0.2)
	Part-time	110 (13.7)	
Work scheduling	Non-shift work	151 (18.8)	2 (0.2)
	Shift work (with night shifts)	502 (62.5)	
	Shift work (without night shifts)	150 (18.7)	

**Table 2 ijerph-19-09541-t002:** Summary of the results of the Confirmatory Factor Analysis of the Maslach Burnout Inventory performed using the sample from the pre-COVID-19 wave (n = 740).

Structure	Modifications with Respect to the Original Structure	CFI	RMSEA	Issues Affecting the Fit of the Structure *
Original three-correlated factors structure [24,26]		0.8084	0.0829	Presence of cross-loadings of items 6 (EE and DEP), 12 (EE, PA) and 16 (EE, DEP). Presence of correlated errors between items 6 and 16, 1 and 2, 2 and 3.
Three-correlated factors structure, with deletion of items 12 and 16	Removed items 12 and 16	0.8639	0.0730	Presence of cross-loadings of items 6 (EE and DEP) and 2 (EE, DEP). Presence of local dependencies between items 1 and 2, 2 and 3.
Previous structure with further deletion of item 2, similarly to Kim and Ji, 2009 [47]	Removed items 2, 12, 16	0.8902	0.0639	Presence of cross-loadings of items 6 (EE and DEP) and 2 (EE, DEP). Presence of local dependencies between items 6 and 5.
Previous structure with further deletion of item 6, similarly to Kanste et al., 2006 [48]	Removed items 2, 6, 12, 16	0.9260	0.0532	

Note. * Issues affecting the fit of the examined structures were detected based on modification indices. Abbreviations: EE = Emotional Exhaustion; DEP = Depersonalization; PA = Personal Accomplishment.

**Table 3 ijerph-19-09541-t003:** Summary of the results of the Multigroup CFA performed to assess the measurement invariance of the Maslach Burnout inventory across study waves (n = 369).

Type of Invariance	CFI	RMSEA	ΔCFI	ΔRMSEA	Comment
Configural invariance	0.9140	0.0457			Configural invariance was met
Metric invariance	0.9075	0.0467	0.0065	0.001	Metric invariance was met
Scalar invariance	0.8440	0.0597	0.0635	0.013	Complete scalar invariance was not met
Scalar invariance (freed the constraints to the intercepts of items 4, 8, 13, 14 and 20)	0.8836	0.0518	0.0239 *	0.005 *	After freeing the loadings of five items, partial invariance was not met

Note. * Calculated using the fit indices of the structure employed to assess metric invariance as reference.

## Data Availability

Anonymized study data are available upon motivated and reasonable request to the corresponding author.

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
