# Peer review of "The North Italian Longitudinal Study Assessing the Mental Health Effects of SARS-CoV-2 Pandemic Health Care Workers—Part II: Structural Validity of Scales Assessing Mental Health"

_ijerph, 2022, doi:10.3390/ijerph19159541_

Round 1
Reviewer 1 Report
Dear authors,
thank you for considering that Covid 19 may have implications for the factor structure of a number of well-known instruments.
However,
it is unfortunate that the details regarding the method need to be read from paper 22, which is not publicly available. How to assess the quality of method and sample when this information is missing?
Furthermore, changing an existing and well-tested scale based (MBI) on one data set is a bit risky. I would expect a more through proof that the scale needs to be adjusted. Also, quite a nyumber of respondents from the first wave did not respond the secondd time. But with an anonymous sample, it is not possible to link respondents from both waves, that does not improve the reliability of the second wave. I would consider this to be but the first indication, not a final proof. The analysis itself is fine, but changingexisting instruments should always be researched carefully, I would say that your conclusions are rather too confident.
However,
Reviewer 2 Report
I have read the article carefully, although it is part of a larger study I think it is interesting and will be of help to readers even at an international level, it deserves to be published.
I only have a few comments.
I believe that the authors could also insert the limits of the present study not just the limits of the external questionnaires.
A brief reference to the Italian legislation for the protection of workers in the covid era could be included (DOI: 10.3390 / healthcare9010017)
Authors should review editorial rules for citations and imagery.
